# Preparation and Characterization of Nanocomposite Hydrogels Based on Self-Assembling Collagen and Cellulose Nanocrystals

**DOI:** 10.3390/polym15051308

**Published:** 2023-03-05

**Authors:** Ya Li, Xiaotong Dong, Lihui Yao, Yajuan Wang, Linghui Wang, Zhiqiang Jiang, Dan Qiu

**Affiliations:** 1School of Materials and Chemical Engineering, Ningbo University of Technology, Ningbo 315211, China; 2Zhejiang Institute of Tianjin University, Ningbo 315201, China; 3College of Food and Pharmaceutical Sciences, Ningbo University, Ningbo 315800, China

**Keywords:** cellulose nanocrystal, collagen, self-assembly, nanocomposite hydrogel

## Abstract

Collagen (Col) hydrogels are an important biomaterial with many applications in the biomedical sector. However, deficiencies, including insufficient mechanical properties and a rapid rate of biodegradation, hamper their application. In this work, nanocomposite hydrogels were prepared by combining a cellulose nanocrystal (CNC) with Col without any chemical modification. The high-pressure, homogenized CNC matrix acts as nuclei in the collagen’s self-aggregation process. The obtained CNC/Col hydrogels were characterized in terms of their morphology, mechanical and thermal properties and structure by SEM, rotational rheometer, DSC and FTIR, respectively. Ultraviolet-visible spectroscopy was used to characterize the self-assembling phase behavior of the CNC/Col hydrogels. The results showed an accelerated assembling rate with the increasing loading of CNC. The triple-helix structure of the collagen was preserved with a dosage of CNC of up to 15 wt%. The CNC/Col hydrogels demonstrated an improvement in both the storage modulus and thermal stability which is attributed to the interaction between the CNC and collagen by the hydrogen bonds.

## 1. Introduction

Type I collagen is the most abundant protein in mammals and is particularly provided in the form of fibrils and fibers. The major function of collagen is to provide shape and mechanical strength. The latter is achieved by the intermolecular crosslinking of the collagen molecules in supramolecular aggregation [1]. All collagen proteins possess the basic triple helix, which is based on multiple repeats of the simple tripeptide Gly-X-Y in which X and Y are most commonly occupied by proline and hydroxyproline, respectively [1,2,3]. The triple helices self-assemble to form fibrils on the nano scale [4]. Further linear entwining of the nanofibrils gives rise to microfibrils, which can further assemble into large fibrillar structures [5,6,7,8]. Collagen self-assembly fibrils can be observed in vivo as well as in vitro under certain conditions [9]. Due to their low immunogenicity, controlled biodegradability, ability to promote cell growth and good biocompatibility, collagen-based hydrogels have been widely used in biomedical applications [3]. However, when compared with synthetic polymers, collagen-based hydrogels usually have poor mechanical properties and are limited in their application as wound dressings and scaffolds for tissue engineering [10]. To improve the mechanical properties and reduce the biodegradation rate of collagen-based hydrogels, collagen is generally chemically crosslinked by crosslinking agents, such as glutaraldehydes and carbodiimides, or by enzyme catalysis [11]. However, the normal chemical crosslinkers may cause cytotoxicity or tissue calcification. Compared with synthetic crosslinking agents, natural macromolecules demonstrate a low toxicity and good biocompatibility with collagen. Therefore, an increasing number of studies have prepared composite hydrogels by incorporating biopolymers such as chitosan and cellulose [10,11,12,13,14].

Cellulose nanocrystals (CNC) are rod-like crystalline nanoparticles obtained from the controlled acid hydrolysis of cellulose fibers, which were assumed to be ideal reinforcing nanoparticles for materials [14,15]. Due to their nanostructure, CNCs have a high specific surface area and high surface reactivity, and CNCs also have the same molecular structure of cellulose with active hydroxyl groups that can interact with the active groups of collagen to form crosslinking bonds. These properties are the natural advantages of a CNC in the reinforcement reaction with collagen hydrogels. Several composites that incorporate a CNC with collagen or protein have been described. For example, Stein [16] demonstrated a one-pot, liquid-phase method of generating centimeter-scale films of fully aligned collagen fibrils by the addition of nanocrystalline methylcellulose to a collagen solution prior to gelation. Zhang [17] developed a biodegradable gelatin microsphere incorporating porous collagen/CNC scaffold with augmented cell proliferation that was capable of releasing growth factors. Coradin [18] prepared CNC fibrin nanocomposite hydrogels from concentrated fibrinogen solutions with improved mechanical stability as potential cellular scaffolds. Qi [19] reported a new class of CNC/collagen hydrogels with an anisotropic, porous structure and a horizontally tubular array. CNCs can affect the creep and stress relaxation behaviors of hydrogels. However, most of the methods reported focused on chemical modifications of CNC or collagen which had some difficulty in scaling up in vitro and lacked the ease of a one-pot reaction. Additionally, these reported works did not take the self-assembling ability of collagen into consideration.

In this work, we attempted to obtain collagen-based hydrogels produced with nanomaterials from natural polymers. We provide a one-step method of preparing nanocomposite hydrogels based on collagen self-assembly in vitro by integrating collagen molecules with a CNC matrix. To achieve a homogenous and orientated CNC matrix suspension, CNCs were first treated with a high-pressure homogenizer without any chemical treatment [20]. The effect of the CNC on collagen self-assembly kinetics was explored by UV-vis. The morphology and properties of the CNC/Col hydrogels were characterized using SEM, DSC and rotational rheometer, respectively. The interaction between CNC and collagen was analyzed using XPS. The triple-helix structure of collagen in CNC/Col hydrogels was characterized by FTIR.

## 2. Materials and Methods

### 2.1. Materials

The CNC was processed from commercial medical absorbent cotton. The acetic acid solution of Type I rat tail collagen (5 mg/mL) was purchased from Solarbio, China. Collagen powder from pollock skin was purchased from Macklin, China. The other normal chemicals, such as H_2_SO_4_ (98%), NaOH and a PBS buffer solution (pH 7.4), were all analytical pure and were commercially available from SCRC Inc. Shanghai, China. All the chemicals were used as received.

### 2.2. Preparation of CNC and CNC/Col Nanocomposite Hydrogels

The preparation of CNC was based on the acid hydrolysis method [21]. Absorbent cotton was treated with acetic acid (1% *w*/*v*) and NaClO (1.2% *w*/*v*) solution at 75 °C for 2 h to remove lignin. After being washed with distilled water, the sample was treated with NaOH solution (2% *w*/*v*) for 12 h at 90 °C to remove hemicellulose. After washing and drying in an oven, the mercerized cotton was hydrolyzed with 64% *v*/*v* sulfuric acid at 45 °C under stirring for 4 h. After hydrolysis, the suspension was diluted with cold water (4 °C) and washed by repeated centrifugation until a colloidal suspension was obtained. The suspension was dialyzed using dialysis membranes with 12–14 kDa molecular weight against ultra-pure water until a neutral pH was reached. The final dispersion was sonicated for 15 min and stored at 4 °C for use.

The resulting thick, milky white suspension of CNC was used in the preparation of CNC/Col hydrogels. A total of 5, 10, 15 and 20 wt% of CNC based on the collagen mass were dispersed in phosphate buffer solution (PBS) and then homogenized three times at a pressure of 20,000 psi, using a NanoGenizer 30 K high-pressure homogenizer. The dosages of CNC refer to the mass percentage of solid CNC without water to collagen.

Fish collagen was added to a flask that contained various volumes of high-pressure-treated CNC suspension, and the mixed solution was then adjusted to a concentration of 1 mg/mL (calculated on collagen) under magnetic stirring. The flask was placed into a constant-temperature bath at 37 °C at the desired time for collagen self-assembly to obtain CNC/Col nanocomposite hydrogels. Some of the hydrogels were stored at 4 °C for rheology measurements. After the fully assembly of collagen, the hydrogels were immersed in aqueous formaldehyde (37 wt%) for 4 h and then freeze-dried for further characterization.

### 2.3. Characterization

The self-assembling kinetics of CNC/Col were characterized by ultraviolet-visible spectroscopy (UV-vis) (UV2600i, Shimadzu, Kyoto, Japan). UV-vis absorption spectra of the PBS solution with a collagen concentration of 1 mg/mL and various loadings of CNC were collected at an interval of 5 to 60 min at 220 nm. The structural features and morphologies of the lyophilized BC/Col hydrogel were studied using SEM (S-4800, Hitachi, Tokyo, Japan) with an accelerating voltage of 5 kV. All samples were coated with gold for 20 s at 50 mA to obtain a conductive coating and to reduce electron charging and sample burning. To observe the morphology of the collagen fibers formed in the self-assembling process, after self-assembly for 10 to 60 min a 100 μL solution of BC/Col was coated on mica surface and rinsed three times to wash the salt. SEM images of the BC/Col hydrogels on mica were taken. The diameter of the collagen fibrils was calculated by a colored pathology image text analysis system, V3.0 software. Type I rat tail collagen was used here because of its easy gelation property. The changes in the protein backbone (the triple-helical part of the collagen) were characterized by Fourier-transform infrared spectroscopy (FTIR). The samples were blended with KBr and analyzed by FTIR (Nicolet is10, Thermo Scientific, Waltham, MA, USA) at a resolution of 4 cm^−1^ in the range of 400–4000 cm^−1^. The elemental composition and chemical structure change of the CNC, Col and CNC/Col hydrogels were determined by an X-ray photoelectron spectroscopy (XPS) (Nexsa XPS, Thermo Scientific, MA, USA). The XPS survey was collected at pass energy of 100 eV with a scan step of 1.0 eV. The O1s and C1s spectra were collected at pass energy of 30 eV with a scan step of 0.1 eV. The X-ray diffraction patterns of the samples were collected with an X-ray diffraction instrument (XRD) (D8 advance, Bruker, Berlin, Gemany), using a Cu Kα radiation at a wavelength of 0.154 nm from 2θ = 5° to 40° at a scan rate of 4°/min. A thermal analysis was carried out using a differential scanning calorimetric apparatus (DSC 214 polyma, Netzsch, Waldkraiburg, Germany). Under nitrogen flow, the samples were heated from 0 to 200 °C at a rate of 10 °C/min. The rheological behavior of the CNC/Col hydrogels was investigated using a rotational rheometer (DHR-1, TA, Newcastle DE, USA) equipped with a 40 mm diameter plate. The distance between the plates was 2.5 mm. A frequency sweep from 0.01 to 20 Hz was used to record the storage modulus (G′) and loss modulus (G″) of the CNC/Col hydrogel samples at a temperature of 25 °C. 

## 3. Results and Discussion

### 3.1. Self-Assembly Kinetics of CNC/Col Hydrogels

The self-assembly phase of collagen from the 1 mg/mL CNC/Col composite solution at pH 7.4 and an incubation temperature of 37 °C was measured by UV absorbance changes using a UV-vis spectrophotometer. Figure 1 shows the curves recorded the UV absorbance of the solution with different dosages of CNC. All curves show a sigmoid profile with three distinct phases that are typical for filamentous protein aggregation, according to the literature [22]. The first step is a lag phase during which the absorbance does not change and the fibril nucleation begins. The second step is a growth phase with a rapid absorbance change that corresponds to the core growth of fibrils. In the third step, the absorbance tends to have a constant value, including the formation of three-dimensional networks [1,22,23]. As shown in Figure 1, the self-assembly process of pure collagen has the longest lag phase and shortest growth phase: nucleation finished in 27 min and the aggregation formed in 120 min. With the addition of 5 wt% CNC, both the lag phase and the growth phase accelerated to 15 and 30 min respectively, and aggregation finished in almost a quarter of the time taken for neat collagen, indicating that the addition of CNC strongly influenced the collagen fibril formation process and led to an accelerated assembly rate of collagen. With a further increasing dosage of CNC of up to 20 wt%, the self-assembly rate leveled off when compared with that of the 15 wt%.

The self-assembly process of collagen in the presence of CNC is illustrated by Figure 1. During the lag phase, collagen molecules are associated with CNC by hydrogen bonds to form metastable nuclei. Microfibrillar aggregation starts from the nuclei with further collagen molecules accreting during the growth phase. Neighboring microfibrils are interdigitated with one another, forming bundles of fibrils. Continuing growth in fibril length and width finally formed hydrogels with a double-network structure of collagen fibrils intercrossed with the CNC matrix. For pure collagen, small numbers of collagen molecules are first associated to form small segments and act as metastable nuclei upon which molecules aggregate during the growth phase [24]. For CNC/Col hydrogels, the addition of CNC acts as a nucleation agent the same way that it acts in the crystallization behavior of some macromolecules. The collagen molecules can begin aggregating on the CNC nuclei, which accelerates the lag phase and the growth phase of collagen self-assembly [25,26].

### 3.2. Morphology of CNC/Col Hydrogels

Figure 2 shows the morphology of the collagen fibrils formed in the CNC/Col hydrogel assembly process. After 10 min of self-assembling, a monolayer of single collagen fibrils with right-handed helices was observed, which was similar to other Type I collagen arranged on a mica or PS surface [27,28,29]. The fibrils had widths from 100 to 250 nm and fiber lengths of tens of microns (Figure 2a). Collagen fibers aggregated with the evolution of the self-assembly and the fibrils chaotically intertwined with each other, forming a network structure with fiber bundles stacked layer by layer (Figure 2b). When the collagen fibers assemble and densify at an increasing aggregation level, a collagen-based hydrogel is obtained. The CNC/Col hydrogel displayed a porous structure with randomly disposed fiber arrangements in a dense network (Figure 2c). As it contributes to the supermolecular organization of collagen and the interconnected pores in three-dimensional directions throughout the hydrogel, the CNC/Col hydrogel could be applied in many fields, including biomaterials and nanotechnologies [3]. 

Figure 3 shows the morphology of the CNC/Col freeze-dried hydrogel prepared from fish collagen with various dosages of CNC. The pure collagen hydrogel showed a loose and porous structure with holes of several microns (Figure 3a). The CNC/Col hydrogels with CNC loadings of 5 wt% and 10 wt% also presented a interconnected porous structure with numerous pores inside, and the pore size decreased (Figure 3b,c). With a dosage of CNC of up to 15 and 20 wt%, the porosity decreased and the hydrogel structure became dense and compact, and a smooth surface could be observed (Figure 3d,e). This can be attributed to the presence of CNC in collagen aggregation and gelation. With the increasing addition of CNC, the aggregation behavior became more prominent, giving rise to the formation of a larger bundle. More joint points formed when the amount of CNC increased, and collagen fibrils interlaced closely, resulting in the compact structure of the CNC/Col hydrogels [11,16]. In addition, with the dosage of CNC rising up to 20 wt%, the denaturation of collagen to gelatin may also have contributed to the dense morphology of the hydrogels. The denaturation of the collagen was proved in the following FTIR analysis.

### 3.3. Intermolecular Interaction between Col and CNC

The relative concentrations of the C and O atoms at surfaces of CNC, Col and CNC/Col hydrogels were determined by XPS analysis. Peaks of C, N and O elements can be observed from the wide scan spectra of the samples in Figure 4a. As shown in the inset table in Figure 4a, the percentages of C and O atoms of collagen were 49.2 and 50.8%, respectively, and the ratio of C to O was approximately 0.97. For the CNC/Col hydrogel, the ratio of C to O increased to 1.26 due to the higher amount of C to O in CNC [30].

Chemical states can be obtained from the high-resolution O 1s peak in the XPS spectrum. The O 1s XPS spectrum of Col and CNC/Col is shown in Figure 4b,c, and the calculated data are shown in Table 1. The big O 1s peak fits into two peaks which correspond to various chemical shifts of oxygen. For the Col sample, O 1s1 and O 1s2 can be assigned to the peaks at 531.77 ev (-C=O) and 533.06 ev (C-O-H) (Table 1) [31]. After the self-assembly of collagen in the CNC/Col hydrogel, the binding energy of O 1s1 and O 1s2 decreased by 0.07 and 0.48 ev, respectively, and the relative content of O 1s2 (C-O-H) had an increase of 16.82% (Table 1). The electron-withdrawing property of negative oxygen ions resulted in a lower binding energy. Therefore, the content of O 1s2 (C-O-H) rose [32]. This may indicate a hydrogen bond formation between the free hydroxyl group of CNC and the carbonyl group of Col. With the increasing dosage of CNC, the stronger interaction between the CNC and Col in hydrogen bonds may cause the change of collagen conformation.

### 3.4. FTIR Spectra of CNC/Col Hydrogels

Figure 5 shows the FTIR spectra of the Col and CNC/Col hydrogels. The triple-helical structure of collagen can be characterized by FTIR spectra located in the so-called Amide A, Amide I, and Amide II bands [11]. As shown in Figure 5, the amide A band appears at approximately 3333 cm^−1^ and is associated with N-H stretching vibration. Its frequency depends on the hydrogen bond and the degree of crosslinking. The band at 1646 cm^−1^ is the typical peak of amide I, which is associated mainly with the stretching vibration of C=O groups of the collagen helical polypeptide backbone. The intensity of amide I is considered a sensitive marker of changes in the peptide chain conformation. The amide II band appears at 1553 cm^−1^ and is attributed to the coupling of the N-H bending vibration and C-N stretching vibration in the amide group. It was found that the amide II band is also sensitive to the conformational change of peptide; however, the correlation is less straightforward than for the amide I band. The amide band appears at approximately 1240 cm^−1^ and is associated with the C-O stretching vibration. The band at 1450 cm^−1^ is attributed to CH_3_ asymmetric bending vibration. The ratio of the absorbance of Amide III to that of 1450 cm^−1^ peak indicates the integrity of the triple-helix structure of collagen. If the ratio is far less than 1.0, it shows the loss of the structure. The difference of the frequencies between amide I and amide II (∆υ) indicates the degradation of the collagen. If the difference is close to 100 cm^−1^, it indicates the preservation of the integrity of the structure [10]. The Amide III/A_1450_ ratio and the band frequency ∆υ are summarized in Table 2. 

The ratio of Amide III/A_1450_ and the frequency difference, ∆υ, decreased slightly after composition with CNC (Table 2). Up to the dosage of 15%, the ratio was higher than 1.0 and the ∆υ was very close to 100 cm^−1^, indicating the integrity of collagen triple-helix structure. When the dosage reached 20%, the ratio decreased to 0.95 and ∆υ was a little higher than 100 cm^−1^, indicating the loss of the collagen’s triple-helical structure. In addition, with the increasing dosage of CNC, the intensity of amide I and amide II demonstrated a small decrease and the peak of amide III was very weak (Figure 5). This reveals that the collagen triple helical structure was maintained after being combined with CNC. However, for a higher dosage of CNC (20%), the stronger correlations of CNC with the collagen side chains may cause greater disorder in arrangement of collagen, implying the loss of the triple-helical structure and a reduction in the molecular arrangement [11]. 

### 3.5. XRD Analysis of CNC/Col Hydrogels

The XRD patterns for pure Col and CNC/Col hydrogels are shown in Figure 6. For pure collagen, no typical diffraction peak was observed, proving that the Col is quite an amorphous polymer. The XRD patterns of the CNC/Col hydrogels with CNC loading from 5 wt% to 20 wt% showed characteristic peaks at 2θ = 15.0° and 22.8°. This pattern presents a typical crystalline structure of cellulose I, and the two peaks can be attributed to the cellulose I_α_ and I_β_ phases, respectively [33]. A small shift could be found of the typical peak from 23.0° to 22.8° as the dosage of CNC increased from 5 to 10 wt%. There was then no change of the peak position for 15 and 20 wt% (22.8°), indicating that the CNC crystalline structure was retained in the CNC/Col hydrogels. Moreover, that the intensity of the CNC peaks was enhanced may be attributed to the increasing content of CNC and the lower presence of amorphous collagen.

### 3.6. Thermal Properties of CNC/Col Hydrogels

Thermal stability tests measure the temperature at which collagen denatures from its native and hydrogen bonded triple structure into the form of a random coil. The thermal transition curves of the CNC/Col hydrogels are displayed in Figure 7. Endothermic peaks can be observed in the temperature scanning curves, which refer to the thermal denaturation temperature (T_d_) of collagen. The value of T_d_ reflects the transformation from the native triple helix into a random coil structure of collagen, depending on hydrogen bonds and the cross-linking bonds within the collagen. Samples with a higher T_d_ have more hydrogen bonds or/and fewer hydrophobic bonds [34,35]. For biomedical application purposes, thermal stability is important in the sense of maintaining the stability of collagen and influencing the durability of the collagen-based materials [11]. 

As can be observed from Figure 7, when the dosage of CNC increased from 0 to 10 wt%, the value of T_d_ showed an upward tendency from 60.7 °C to 75.5 °C. It then increased mildly when the CNC dosage increased from 10 to 20 wt%. The maximal T_d_ value of the hydrogels was 77.2 °C at 20 wt%, which was 16.5 °C higher than that of native collagen hydrogels, demonstrating an improvement in the thermal stability. This might be due to the interactions of CNC and collagen through hydrogen bonds; with an increase in the CNC dosage, more hydrogen bonds formed between CNC and collagen, and the crosslinked collagen was therefore more stable [35,36]. The morphologies may also contribute to the thermal stability. Upon connecting with CNC, the collagen fibrils formed chaotic entanglement structures (Figure 3). The higher the dosage of CNC, the more compact the arrangement of the collagen fibrils, resulting in the increase of T_d_. 

### 3.7. Mechanical Properties of CNC/Col Hydrogels

During the rheological tests, the storage modulus (G′) and loss modulus (G″) were recorded as functions of angular frequency, which measure the elastically stored energy and the energy lost on application of shearing force, respectively. The rheological results of the CNC/Col hydrogels formed with various loads of CNC are shown in Figure 8. The first observation is that G′ is larger than G″ for both pure collagen and CNC/Col hydrogels, indicating that these hydrogels behaved as elastic solids [35]. The observed results were similar to those of collagen hydrogels or peptide hydrogels [4,37,38]. 

As shown in Figure 8a, the G′ modulus of CNC/Col hydrogels increased with the increasing dosage of CNC. Under the frequency of 10 rad s^−1^, the G′ of pure collagen gel was 86.9 Pa and the G′ of the hydrogel with CNC of 15% was 790.4 Pa, almost ten times greater. The interactions between CNC and collagen with hydrogen bonds that act as physical cross-links formed a cross-linked network in the hydrogels, leading to the increasing mechanical properties of CNC/Col hydrogels [38]. The increased G′ also reflects increasing collagen crosslinking density. The loss modulus G″ also demonstrated a small increase with increasing loading of CNC, and it was insensitive to frequency. No cross-over of G″ at low frequencies was observed, which is a characteristic of a cross-linked network (Figure 8b) [37].

## 4. Conclusions

In this work, CNC-reinforced nanocomposite hydrogels based on collagen self-assembly were obtained. CNC has a strong influence on the collagen fibril hydrogel formation in terms of both of kinetics and structure. The self-assembly progress was almost four times accelerated in both the lag phase and the growth phase with the addition of 5 wt% CNC. Hydrogels with an interconnected, porous structure were formed with CNC loadings of 5, 10 and 20 wt%. The triple-helix structure of collagen was preserved after combination with CNC; however, the conformational change occurred at a dosage of 20 wt%. With an increasing dosage, a significant enhancement of the thermal stability of CNC/Col hydrogels was obtained, owing to the cross-linking between CNC and Col by hydrogen bonds. The thermal denaturation temperature (T_d_) demonstrated an increase of 16.5 °C. The rheological tests confirmed the elastic character of CNC/Col hydrogels and showed an increase of almost ten times in the storage modulus. XRD results indicated that the crystal structure of CNC was not changed in CNC/Col hydrogels. 

In this context, a biodegradable CNC without any chemical modification was employed in the collagen-based hydrogel preparation, acting as nuclei in the collagen-self-assembly process. The experimental results present the hydrogels a data basis for potential applications as biomedical materials. Further experiments on the prepared hydrogels’ biocompatibility and biodegradability will be continued.

## Data Availability

Not applicable.

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
