# Peer review of "Preparation and Characterization of Nanocomposite Hydrogels Based on Self-Assembling Collagen and Cellulose Nanocrystals"

_polymers, 2023, doi:10.3390/polym15051308_

Round 1
Reviewer 1 Report
The work entitled “Nanocmoposite hydrogels bio-based self-assembly collagen and cellulose nanocrystals” report on the improvement of collagen hydrogels’ properties via double-network nanocomposite hydrogels of cellulose nanocrystals (CNC) matrix incorporated with a self-assembled collagen network. They demonstrated that assembly of collagen accelerates with the presence of CNC, promoting the formation of a fibrous network of great potential and improved thermal and mechanical features. Even though the work is of interest, it is poorly written, and many English writing mistakes can be detected along the manuscript. Also, some times the ideas the authors aim to convey do not come through because of that, so English must be improved.
Additionally, the title should be corrected. It is also incomprehensible in the way it is presented at the moment.
Author Response
Thanks for the positive feedback on the revision and the opportunity to publish our research. It is extremely helpful in providing constructive suggestions. We are truly grateful for your excellent advice we have received. Thank you very much!
The english writing was tried to be improved. And the title has been revised as "Preparation and characterization of nanocmoposite hydrogels based on self-assembly collagen and cellulose nanocrystals". All revisions are marked in red in the text.
Reviewer 2 Report
The manuscript "Nanocmoposite hydrogels bio-based self-assembly collagen and cellulose nanocrystals" aimed to develop double-network nanocmoposite hydrogels of cellulose nanocrystals (CNC) matrix incorprated with Col network based on collagen self-assembling were prepared to improve Col hydrogel’s properties. The idea seems to be novel and the experiments were carried out nicely. However, the manuscript has shortcomings that prevent its publication in the current form.
More significant findings need to be included in the abstract. There are also some grammatical mistakes in the abstract.
The introduction part needs to be elaborated and includes more details. As well as more literature review needs to be added to the introduction part. I suggest these references to be included in this part:
10.3390/molecules28010306
10.3390/ma15217663
The results need to be discussed in more depth and compare with the gold standard.
The conclusion part need to be elaborated and to include author's perspectives and opinions.
The quality of figures 4 and 7 need to be improved.
Author Response
Thanks for the positive feedback on the revision and the opportunity to publish our research. It is extremely helpful in providing constructive suggestions. We are truly grateful for your excellent advice we have received. We have carefully analyzed all of your suggestions and indicated the changes we have made in the manuscript.
(1) The abstract was revised and revision details were marked in red color in the text.
(2) The suggested reference paper is very helpful and is added in the reference list.
(3)The results discussion and conclusions were revised as suggested, details refer to the attachment manuscript.
(4)The quality of all the images were improved and changed by color ones.

Reviewer 3 Report
This manuscript by Li et. al, titled “Nanocmoposite hydrogels bio-based self-assembly collagen and cellulose nanocrystals” presented a very detailed study for using Cellulose Nanocrystals in Collagen for forming self assembled hydrogels. Interestingly, increase in mechanical strength of the hydrogel has been presented in this work along with some other phenomenon. However, the manuscript lacks a few important points that have been mentioned below:
1. Title: The authors will need to fix the typo in the title. It also looks very jumbled as a sentence. It will need to be fixed.
2. Minor grammatical errors needs to be checked for some parts of the manuscript. For example, Page 1, line 9, "...odegradation hampered their application": It will be "hampers". Page 2, Line 47, "...CNCs can affect the creep and stress relaxation behaviors of the hydrogels.": This should be new sentence. Page 2, line 56, "The collagen fibirls network formed by self-assembling...": "were" missing from the sentence and it should be "fibrils". Authors need to fix the similar issues in the rest of the manuscript.
3. Page 2, line 64, "... was purchased from Solarbio and fish collagen was purchased from Macklin.": Details of the suppliers need to be mentioned like the details have been mentioned for all the instruments used in this work. This helps in reproducing the work using the same source of materials, if needed. The concentration and supplier details of H2SO4, NaOH and PBS also needs to be mentioned.
4. Page 2, line 69: Typo in the formula of "NaClO" needs to be fixed.
5. Page 2, line 79, "5 to 20 wt% of CNC based on collagen mass was dispersed...": Authors need to mention clearly that 0, 5, 10, 15 and 20% CNC was used. 5-20% would not mean the absolute values within that range.
6. Page 4, line 161, " The fibrils have widths from 100 to 250 nm and fibre lengths...": It needs to be included how these widths were measured. Mentioning the details of software or other techniques used, will reduce confusion.
7. Page 5, Figure 2 and 3: Decimals in the scale of the SEM images are hardly visible and needs to be zoomed in largely. Clear scales need to be presented in the images.
8. Page 6, line 217, "It could be found that there is no significant change of the...": Peak shifts can be observed for 15 and 20% compositions which needs to be mentioned with possible causes.
9. Are there any particular reasons why some of the figures are in black and white while some are in colors? Colored representation would be very helpful for the other plots, especially for Figure 7, similar to Figure 1.
10. References need to be in the same format. For example, Ref 8 does not have bolded year like the rest of the references. Ref 12 is in a different format. Similar errors needs to be revised.
Author Response
Thanks for the positive feedback on the revision and the opportunity to publish our research. It is extremely helpful in providing constructive suggestions. We have carefully analyzed all of your suggestions and indicated the changes we have made in the following response.
- The title has been revised as "Preparation and characterization of nanocmoposite hydrogels based on self-assembly collagen and cellulose nanocrystals".
- All the manuscript was checked and the grammatical errors were tried to be corrected.
- Details of the materials and suppliers was mentioned.
- "NaCLO " was revised to "NaClO".
- was revised as suggested.
- Software information was mentioned.
- All the SEM images were replaced with clear scales.
- Actually, the shift is very small ( about 0.2︒), however it was also mentioned in the revised manuscript.
- All the images were replaced by color ones.
- All references were checked and revised.
